# Newborn Screening for X-Linked Adrenoleukodystrophy: Review of Data and Outcomes in Pennsylvania

**DOI:** 10.3390/ijns8020024

**Published:** 2022-03-23

**Authors:** Jessica R. C. Priestley, Laura A. Adang, Sarah Drewes Williams, Uta Lichter-Konecki, Caitlin Menello, Nicole M. Engelhardt, James C. DiPerna, Brenda DiBoscio, Rebecca C. Ahrens-Nicklas, Andrew C. Edmondson, Francis Jeshira Reynoso Santos, Can Ficicioglu

**Affiliations:** 1Section of Biochemical Genetics, Division of Genetics, Children’s Hospital of Philadelphia, Philadelphia, PA 19104, USA; priestleyj@chop.edu (J.R.C.P.); menelloc@chop.edu (C.M.); engelhardn@chop.edu (N.M.E.); dibosciob1@chop.edu (B.D.); ahrensnicklasr@chop.edu (R.C.A.-N.); edmondsona@chop.edu (A.C.E.); reynosof@chop.edu (F.J.R.S.); 2Division of Neurology, Children’s Hospital of Philadelphia, Philadelphia, PA 19104, USA; adangl@chop.edu; 3Division of Genetic and Genomic Medicine, University of Pittsburgh Medical Center Children’s Hospital of Pittsburgh, Pittsburgh, PA 15224, USA; sarah.drewes@chp.edu (S.D.W.); uta.lichterkonecki@chp.edu (U.L.-K.); 4PerkinElmer, Mass Spectroscopy Unit, Pittsburgh, PA 15275, USA; james.diperna@perkinelmer.com; 5Department of Pediatrics, Perelman School of Medicine, University of Pennsylvania, Philadelphia, PA 19104, USA

**Keywords:** X-linked adrenoleukodystrophy, X-ALD, C26:0, *ABCD1*, newborn screening

## Abstract

X-linked adrenoleukodystrophy (X-ALD) is the most common peroxisomal disorder. It results from pathogenic variants in *ABCD1*, which encodes the peroxisomal very-long-chain fatty acid transporter, causing a spectrum of neurodegenerative phenotypes. The childhood cerebral form of the disease is particularly devastating. Early diagnosis and intervention improve outcomes. Because newborn screening facilitates identification of at-risk individuals during their asymptomatic period, X-ALD was added to the Pennsylvania newborn screening program in 2017. We analyzed outcomes from the first four years of X-ALD newborn screening, which employed a two-tier approach and reflexive *ABCD1* sequencing. There were 51 positive screens with elevated C26:0-lysophosphatidylcholine on second-tier screening. *ABCD1* sequencing identified 21 hemizygous males and 24 heterozygous females, and clinical follow up identified four patients with peroxisomal biogenesis disorders. There were two false-positive cases and one false-negative case. Three unscreened individuals, two of whom were symptomatic, were diagnosed following their young siblings’ newborn screening results. Combined with experiences from six other states, this suggests a U.S. incidence of roughly 1 in 10,500, higher than had been previously reported. Many of these infants lack a known family history of X-ALD. Together, these data highlight both the achievements and challenges of newborn screening for X-ALD.

## 1. Introduction

First appreciated through its more severe phenotypes in the 1910s and 1920s, X-linked adrenoleukodystrophy (X-ALD; MIM: 300100) is the most common peroxisomal disorder, affecting about 1 in 15,000 births [1,2]. It is caused by pathogenic variants in the *ABCD1* gene, which encodes an ATP-binding cassette membrane protein that transports very-long-chain saturated fatty acids (≥C22:0; VLCFAs) into the peroxisome for β-oxidation [3,4]. Defects in *ABCD1* result in the accumulation of cholesterol esters containing VLCFAs [5,6,7], which is postulated to result in disease manifestations secondary to the toxicity of VLCFAs through the generation of oxidative stress, inflammation, and generalized peroxisomal dysfunction [8]. More than 900 *ABCD1* variants have been identified and are cataloged in the ALD Variant Database available online at https://adrenoleukodystrophy.info/mutations-and-variants-in-abcd1 accessed on 15 March 2022 [4].

The accumulation of VLCFAs in the central nervous system and adrenal cortex results in a disease spectrum in affected males [9,10,11,12]. There is a broad spectrum of neurologic disability associated with X-ALD and no known genotypic–phenotypic correlation [4,13,14]. The most severe and earliest onset form is the childhood cerebral (CCALD) subtype. CCALD is most commonly described in school-age boys who present with progressive behavioral changes, followed by rapid motor and cognitive deterioration. Untreated, death occurs within years of symptom onset [15]. There are examples of adult-onset cerebral involvement as well, although the course of these later-onset cases is less well defined. Adrenomyeloneuropathy (AMN) manifests in adulthood predominantly with spinal cord involvement, including progressive spasticity, weakness, neuropathies, and loss of bowel/bladder control. This form is of variable severity and can affect women as well. In addition to the neurologic features, many males are affected by primary adrenal insufficiency (Addison disease) [16]. Untreated, adrenal insufficiency can affect growth and can be fatal during periods of physiological stress.

Although X-ALD is classically considered a disease of males, many females demonstrate often under-recognized clinical manifestations. In one study, seven of eight female subjects >60 years of age with *ABCD1* variants manifested symptoms similar to those of AMN (particularly pain, weakness, and fecal incontinence) [17]. In another study, 29 of 33 females with *ABCD1* variants were symptomatic, with an average age of symptom onset of 39 years [18]. Symptom progression is slow [19]. Adrenal insufficiency is uncommon in this population, estimated to occur in 1–2% of individuals [20]. There are rare reports of females with cerebral adrenoleukodystrophy [21] hypothesized to be related to skewed X inactivation [17]. Additionally, their diagnosis can be more challenging, as female patients do not always demonstrate elevations in VLCFAs [17,19]. In one study, 5 of 20 females with X-ALD had a plasma VLCFA level in the normal range [19].

In the 1980s, elevated plasma VLCFAs, particularly hexacosanoate (C26:0), were recognized as biomarkers for patients with X-ALD [22]. Diagnosis of X-ALD today relies on quantitation of elevated plasma VLCFAs (which are not uniformly present in female heterozygotes [18]), coupled with identification of *ABCD1* pathogenic variants [4]. Additionally, the diagnosis of CCALD can be suggested by characteristic magnetic resonance imaging (MRI) findings, including increased T2 intensities in white matter structures of the corpus callosum, pyramidal tracts, and brainstem consistent with demyelination [23]. These can be present up to two years prior to symptom onset [24,25].

In CCALD, hematopoietic stem cell transplantation prevents disease progression and improves both survival and function measures, with the best outcomes achieved when transplant is performed at the earliest signs of MRI abnormality [26,27,28,29]. Management of adrenal insufficiency with hormone replacement is well-established [30]. Emerging therapies, including gene therapy and targeted small molecules, offer hope for additional treatment options for cerebral ALD and AMN in the future [31,32].

Given the benefits of early disease identification for treatment options and disease outcomes for adrenal insufficiency and CCALD phenotypes, X-ALD was targeted for inclusion in newborn screening (NBS) protocols beginning in the early 2000s [2]. In December 2013, New York was the first state to add X-ALD to its NBS portfolio [2]. X-ALD was added to the United States Recommended Universal Screening Panel in 2016 [33,34]. Presently, 21 states and the District of Columbia include the condition in their NBS panels, and it is anticipated that this number will continue to grow.

X-ALD was added to Pennsylvania’s NBS panel in April 2017. Since then, more than 500,000 neonates have been screened for the condition. Here, we detail the Pennsylvania X-ALD NBS experience, review the clinical outcomes, and compare ours with the published experiences of six other states.

## 2. Materials and Methods

### 2.1. Human Subjects Research

The Institutional Review Board at the Children’s Hospital of Philadelphia determined that this study met exemption criteria per 45 CFR 46.104(d) 4(iii). A waiver of HIPAA authorization per 45 CFR 164.512(i)(2)(ii) was granted for accessing identifiable information from medical records. This study utilized deidentified patient data collected by the Pennsylvania Department of Health Newborn Screening Program. A retrospective review of Pennsylvania’s NBS results and outcomes was performed on consecutive filter paper samples from initiation of X-ALD screening on 1 April 2017 through 26 May 2021. Clinical data were gathered via retrospective review of the electronic medical records.

### 2.2. Pennsylvania X-ALD Newborn Screening Protocol

Dried blood spot specimens were collected by birth hospitals at 24–72 h of life for full-term infants and at birth with a repeat filter paper recommended at 24–72 h of life for preterm infants. Specimens were sent to the Pennsylvania Department of Health’s Newborn Screening and Follow-Up Program’s contracted laboratory (PerkinElmer, Waltham, MA, USA) to perform specimen card analysis. The PerkinElmer X-ALD NBS protocol employed a two-tiered testing process to detect X-ALD and eliminate interference occurring at the same mass as C26:0-lysophosphatidylcholine (C26:0-LPC) (Figure 1). The first-tier test was performed via flow injection analysis tandem mass spectrometry (FIA-MS/MS) to measure C26:0-LPC [35]. An 1/8” punched dried blood spot was extracted in methanol at room temperature with a D4-C26:0-LPC internal standard. The extract was diluted with acetonitrile and water plus formic acid. That mixture was analyzed on a tandem mass spectrometer equipped with an electrospray ion source in positive ion mode for flow injection analysis. For newborns with first-tier C26:0-LPC concentrations above the cutoff of 0.36 µmol/L, a second-tier test was performed on the same sample via liquid chromatography tandem mass spectrometry (LC-MS/MS) [36]. This was performed in the manner described above, except a Waters Xterra C8 liquid chromatography column was inserted between the autosampler and mass spectrometer. For newborns with second-tier C26:0-LPC concentrations above the cutoff of 0.15 µmol/L, a second dried blood spot was requested, and only the second-tier C26:0-LPC measurement was performed. If the C26:0-LPC concentration remained above the 0.15 µmol/L cutoff on the repeat specimen, molecular confirmation was performed via next-generation sequencing of the *ABCD1* gene. American College of Medical Genetics standards were used to guide variant interpretation [37].

### 2.3. X-ALD Newborn Screening Follow-Up Care

Infants with elevated C26:0-LPC on repeat dried blood spot specimens (a “positive” NBS) were referred to a biochemical genetic specialist. Selection of a particular referral center was at the discretion of the infant’s primary care provider. The Pennsylvania NBS laboratory also notified the biochemical genetic referral center nearest the family of the positive screening result.

Infants were promptly evaluated by a multidisciplinary biochemical genetics team, including a physician and genetic counselor. At the initial visit, detailed medical and family histories were collected to identify additional at-risk individuals. Social/emotional support was available through genetic counselors or social workers. Confirmatory plasma VLCFAs were obtained. For infants in whom an *ABCD1* variant had been identified, consent was obtained from parents for testing to determine inheritance. Depending on their age, at-risk brothers were also tested for the *ABCD1* variant. For individuals in whom no *ABCD1* variant had been identified or those with concerning signs/symptoms upon evaluation, a peroxisomal biogenesis gene panel was obtained. This included copy number variant analysis of *ABCD1*. Genetic counseling included education about X-ALD and its phenotypes, X-linked inheritance, and recurrence risk. For females with *ABCD1* variants, there was interinstitution variability in follow-up recommendations during the pediatric period. Some referral centers saw female patients at one year of age and as needed during childhood; others did not. Universally, re-evaluation was recommended at 16–18 years of age to discuss reproductive implications. Male patients with confirmed VLCFA elevations, regardless of *ABCD1* variant interpretation, were referred to leukodystrophy subspecialists for co-ordinated X-ALD disease surveillance. Surveillance included regular neurologic and endocrine evaluation beginning at referral, with repeated cerebral magnetic resonance imaging beginning at 12 months of age and adrenal function screening (plasma cortisol and adrenocorticotropic hormone levels) beginning at four months of age, per previously published guidelines [25,38,39].

## 3. Results

### 3.1. Pennsylvania X-ALD Newborn Screening Outcomes

Between 1 April 2017 and 26 May 2021, NBS was performed on 542,554 Pennsylvanian babies (264,224 or 48.7% female; 278,330 or 51.3% male). Of those, 661 babies (0.001%) had initial dried blood spot specimens concerning for X-ALD with C26:0-LPC levels greater than 0.36 µmol/L on first-tier screening and 0.15 µmol/L on second-tier screening. Repeat dried blood spot specimens were requested, of which 610 demonstrated normal C26:0-LPC levels less than 0.15 µmol/L and 51 demonstrated elevated C26:0-LPC levels equal to or exceeding 0.15 µmol/L (Figure 2). These 51 were considered positive screens and referred for biochemical genetic evaluation regardless of *ABCD1* variant identification. Referrals for positive NBS were distributed across five different centers. Within the cohort of screen-positive newborns, gestational ages ranged from 33 to 41 weeks, with a median gestational age at delivery of 39 weeks. Birth weights ranged from 1124–4435 grams, with a median birth weight of 3130 grams. Race as reported on the submitted dried blood spot specimen was 66.7% white, 13.7% black, 10% Asian, 5.9% unknown, and 3.9% other, roughly commensurate with the known demography of Pennsylvania as reported on the United States 2020 census. Clinical data from biochemical genetic center referrals were available for 44 individuals. The outcomes of screen-positive cases are summarized in Figure 2, with patient characteristics and confirmatory testing outcomes detailed in Table 1 for cases with identified *ABCD1* variants and Table 2 for cases without identified *ABCD1* variants.

In total, 21 hemizygous males with X-ALD were identified through PA NBS. Seven harbored pathogenic or likely pathogenic variants in *ABCD1*, all of which were missense or nonsense variants. A total of 23 heterozygous females were identified through PA NBS, among which 16 harbored pathogenic variants, including 15 missense or nonsense variants and one deletion encompassing *ABCD1* exons 3 and 4. There were two false-positive results (Cases 2 and 3 in Table 2). One patient’s parents refused additional evaluation (Case 1 in Table 2). Four children screened positive for X-ALD and were subsequently diagnosed with peroxisomal biogenesis defects (Cases 4–7 in Table 2).

Pennsylvania NBS identified individuals carrying 37 different *ABCD1* variants, of which 21 were pathogenic or likely pathogenic and 16 were variants of uncertain significance (VUS; Table 1). Of the VUSs, 10 had been previously reported in the X-ALD Variant Database [4], most often identified through other newborn screening programs. One VUS (c.1253G>A; p.Arg418Gln) was noted to be present at a frequency of 1 in 183,090 in gnomAD (Case 32 in Figure 1).

We examined the relationship between *ABCD1* variant classification and C26:0 levels. Second-tier screening C26:0-LPC levels were 0.50 ± 0.23 µM for the 7 male patients harboring likely pathogenic/pathogenic variants and 0.37 ± 0.22 µM for the 14 male patients harboring VUSs. This difference was not statistically significant by student’s *t*-test (*p* = 0.21). Results from clinical VLCFA analysis were available for 3/7 male patients harboring likely pathogenic/pathogenic variants and 9/14 male patients harboring VUSs. The average C26:0 levels were 3.09 ± 0.89 µmol/L and 2.59 ± 0.72 µmol/L, respectively. This difference was not statistically significant by student’s *t*-test (*p* = 0.34).

### 3.2. Clinical Assessment and Outcomes

Clinical data were available for 16 of the 21 male patients with *ABCD1* variants. The oldest patient at the time of last neurologic follow-up was three years nine months of age (Case 42 in Table 1). Of those 16 boys, none had MRI findings suggestive of CCALD and none had undergone hematopoietic stem cell transplantation. Four patients had laboratory testing showing adrenal insufficiency. Three boys had been receiving treatment for their adrenal insufficiency since the ages of 2, 4, and 12 months (Cases 24, 26, and 42 in Table 1, respectively). One boy (Case 25 in Table 1) was pending additional testing at 14 months of age prior to initiation of corticosteroid replacement therapy.

Inheritance was assessed via cascade testing of the family members of infants with *ABCD1* variants, and results were available for 34/44 infants (77%; Table 1). Whereas most variants were maternally inherited (28/34; 82%), one pathogenic variant identified in a female patient was inherited from her asymptomatic father (Case 2 in Table 1), and three females carried de novo pathogenic variants (Cases 5, 7, and 10 in Table 1). Additionally, for two female patients, maternal testing was negative, but paternal testing was not available (Cases 9 and 12 in Table 1).

A detailed family history of infants with elevated C26:0-LPC on NBS was collected during their initial clinical evaluation prior to familial cascade testing. This information was available for 38/44 cases (86%). There was a known family history of X-ALD disease phenotypes for 7/38 cases (18%). Of those, six cases harbored “likely pathogenic”/“pathogenic” variants, and two cases harbored VUSs in *ABCD1*. Although there was no history of family members with formal X-ALD diagnoses, 8/38 cases (21%) had family histories concerning for neurological abnormalities that could be consistent with X-ALD phenotypes. Examples include adult female relatives with leg stiffness/weakness or unexplained incontinence and adult male relatives with atypical multiple sclerosis, neuropathy, tremor, or gait abnormalities. In 6/38 cases (16%), there were family members with *ABCD1* variants identified on NBS and/or subsequent cascade testing but no family history of X-ALD disease phenotypes, including in the screen-positive individuals. Two of those, male children from the same family harbor pathogenic *ABCD1* variants. Finally, there was no history of X-ALD diagnosis, positive X-ALD NBS, or suspicious neurological phenotypes in the extended families for the remaining 17/38 cases (45%).

Pennsylvania X-ALD NBS of younger siblings also resulted in the identification of three additional X-ALD cases, each born prior to April 2017. A male, aged six years nine months, was symptomatic (adrenal insufficiency) at the time of biochemical genetic evaluation. Biochemical testing showed elevated C26:0 (1.20 µg/mL; normal: 0.19–0.35 µg/mL). Molecular testing showed the same *ABCD1* pathogenic variant (c.1978C>T; p.Arg660Trp) that was found in his younger sister (Case 5 in Table 1). Clinically, he is being treated for adrenal insufficiency; receives routine MRIs; and is followed by endocrinology, neurology, ophthalmology, psychiatry/psychology, and transplant surgery. Another male, aged 12 years, was also symptomatic (adrenal insufficiency) at the time of initial biochemical genetic evaluation. Biochemical testing showed elevated C26:0 (0.87 µg/mL; normal: 0.19–0.35 µg/mL). Molecular testing showed the same *ABCD1* VUS (c.700C>T; p.Arg234Cys) as that found in his younger brother (Case 39 in Table 1). Although follow-up for this patient has been challenging, involvement by endocrinology, genetics, neurology, and ophthalmology is planned. A final male, aged two years, was asymptomatic at the time of initial biochemical genetic evaluation. Results of biochemical testing were not available, but C26:0 was reportedly elevated. Molecular testing showed the same *ABCD1* pathogenic variant (c.1772G>A; p.Arg591Gln) as that found in his younger brother (Case 7 in Table 1). Later, a second younger brother was born who also screened positive for X-ALD and carries the same *ABCD1* variant (Case 49 in Table 1). All three brothers remain asymptomatic and are followed by endocrinology and neurology for disease surveillance.

Additionally, one male child had a normal Pennsylvania NBS and was later found to carry an *ABCD1* VUS. His initial NBS dried blood specimen had a C26:0-LPC level of 0.17 µM, well below the first-tier screening threshold of 0.36 µM. As a result, second-tier screening was not performed, and a repeat dried blood spot specimen was not requested. He had a younger brother born three years later who had a Pennsylvania NBS concerning for elevated C26:0-LPC and was found to harbor an *ABCD1* VUS (c.86C>A; p.Ala29Asp). This variant has not been reported in the ALD Variant Database [4]. Because the younger brother was born after the conclusion of our study period, his positive screen is not reflected as a case in Table 1. Cascade testing was performed on family members, including the older brother, who was found to harbor the same maternally inherited *ABCD1* VUS. There was no family history of X-ALD diagnosis or suspicious neurologic phenotypes, although the maternal grandfather did have peripheral neuropathy believed to be a consequence of diabetes mellitus type II. The older brother was evaluated by a biochemical geneticist and VLCFAs were obtained. C26:0 was elevated to 1.34 µmol/L (reference range: 0.17–0.73 µmol/L). He is presently 3 years of age, asymptomatic, and followed by neurology and endocrinology for disease surveillance.

### 3.3. Population Genetics and Test Performance

In our population, the incidence of X-ALD was approximately 1 in 13,000 male live births (21 in 278,330), and the incidence of heterozygous females was approximately 1 in 11,000 live births (24 in 264,224). The incidence of other peroxisomal biogenesis disorders was approximately 1 in 136,000 live births (4 in 542,554). The sensitivity of NBS using this two-tier with sequencing approach for X-ALD, carrier status, or peroxisomal biogenesis disorder detection was 98%. The specificity of NBS for X-ALD, carrier status, or peroxisomal biogenesis disorder was >99%. The positive predictive value of an NBS result with elevated C26:0-LPC was 96%. The negative predictive value of a normal NBS was >99%.

### 3.4. Comparison of X-ALD NBS between States

Finally, we compared our X-ALD NBS practices and outcomes with those of six other U.S. states with published experiences: California, Georgia, Illinois, New York, North Carolina, and Minnesota [2,40,41,42,43,44]. Screening practices universally utilized tiered MS/MS approaches to quantify VLCFAs or C26:0-LPC (Table 3). Molecular confirmation with sequencing of the *ABCD1* gene was performed reflexively by the NBS lab for four states (Table 3). Across the seven studies, including the present study, nearly 3.5 million U.S. newborns were screened (Table 4). In total, 156 male hemizygotes, 173 female heterozygotes, and 46 individuals with other genetic disorders, predominantly peroxisomal biogenesis disorders, were identified during the published study periods (Table 4). Taken together, these data suggest a nationwide incidence for X-ALD near 1 in 10,500 live births.

## 4. Discussion

This study summarizes the Pennsylvania experience with NBS for X-ALD since its introduction to the state screening panel in April 2017. Over the ensuing four years, the state’s two-tier screening approach with reflex to *ABCD1* sequencing identified 21 hemizygous males, 24 heterozygous females, and 4 peroxisomal biogenesis disorders. This is the longest reported follow-up period to date. Additionally, X-ALD NBS of younger siblings facilitated diagnosis of symptomatic older siblings in at least two known cases and facilitated early clinical monitoring of another asymptomatic older sibling in at least one additional case.

A younger sibling’s positive screen also facilitated identification of an asymptomatic older brother with normal C26:0 levels on Pennsylvania NBS. To our knowledge, this is the first report of a false-negative X-ALD NBS result in a patient with a demonstrated *ABCD1* variant and elevated C26:0 on confirmatory biochemical testing. As females with *ABCD1* variants can but do not always demonstrate elevated VLCFA levels [18], some newborn screen results with normal C26:0-LPC levels are to be expected in that population. Detection of false-negative cases in males is complicated by variable phenotypes and variable age at disease onset, as well as by the relatively recent addition of X-ALD to most states’ newborn screening programs.

Since 2013, millions of American babies have been screened for X-ALD at birth through NBS protocols. From the seven states for which X-ALD NBS data have been published, 375 infants have been identified with X-ALD, a peroxisomal biogenesis disorder, or another genetic syndrome through NBS. In the present study, most children with X-ALD lacked a family history, suggesting that if not for NBS, they may not have been diagnosed until well after symptom onset [45]. For the subset that will develop CCALD, NBS offers a chance for close neurologic follow-up and therapy at the earliest signs of disease manifestation, when hematopoietic stem cell transplantation is most efficacious in halting progression. Additional work with long follow-up periods will be needed to determine whether NBS truly improves disease outcomes for boys with CCALD.

Previously, extended family screening suggested an X-ALD incidence of 1 in 17,000 births [1]. Incidence rates following implementation of NBS for the condition are higher, ranging from 1 in 4845 births in Minnesota to 1 in 16,200 in Illinois [2,40,41,42,43,44]. Results from Pennsylvania fell within that range and recapitulated a higher incidence rate than previously appreciated. When published results from NBS programs across the country are taken in aggregate, the incidence of X-ALD is about 1 in 10,500 births. This discrepancy may reflect the detection of milder variants. This is speculative, as there are no established genotype–phenotype relationships presently. The individuals with *ABCD1* variants identified via NBS had elevated C26:0 on screening and elevated VLCFAs on confirmatory testing, suggesting their variants are at least impactful enough to elicit a biochemical phenotype. Analysis of NBS C26:0 levels and confirmatory VLCFA testing did not show any differences between males with *ABCD1* likely pathogenic/pathogenic variants and VUSs, adding to the established belief that X-ALD severity and genotype are not necessarily linked. Whereas previous incidence estimations relied on the identification of symptomatic individuals and subsequent testing of relatives, NBS presents an opportunity to study the natural history of people carrying *ABCD1* variants irrespective of symptom development. The higher incidence may also reflect identification of individuals with de novo variants and those without known family histories but who carry pathogenic variants. It was previously reported that 1.7% of X-ALD hemizygotes harbor de novo variants in *ABCD1* [1]. Although we did not identify any de novo *ABCD1* variants in hemizygote males, there were 2/23 (9%) identified among heterozygous females. Larger sample sizes will be needed to determine the true rate of de novo *ABCD1* variants in populations unbiased by phenotypic presentation or relationship to symptomatic individuals.

Detection of *ABCD1* VUS and the spectrum of phenotypic variability within patients harboring known pathogenic variants remain major challenges in NBS for X-ALD. At present, boys identified with elevated C26:0-LPC or VLCFAs and an *ABCD1* VUS are followed closely by multiple subspecialties for monitoring—identical to how boys with known pathogenic variants are managed. Given the discrepancy between incidence figures calculated from data provided by NBS and those calculated from data provided through expanded family screening, it is possible that a proportion of males with *ABCD1* VUS may never develop disease symptoms or may only develop very mild symptoms. This underscores a recognized need for better biochemical or molecular markers and improved prognostication information for these patients, their families, and their care teams.

Although females with *ABCD1* variants usually demonstrate elevations in their C26:0-LPC and long-chain fatty acids, they do not develop CCALD and only rarely develop adrenal insufficiency [17]. Their most common phenotype is late-adult-onset neuropathy for which treatment is lacking. Thus, inclusion of females in newborn screening for X-ALD provides information that will not be clinically relevant until the patient is of reproductive age and/or when symptoms develop decades in the future. Although at risk-family members can be identified through positive female NBS, this information does not directly benefit the tested individual herself. Despite these issues, one prospective survey of families affected by X-ALD showed broad support for X-ALD NBS for both male and female babies (90%) [46]. However, parent perspectives from families unaffected by X-ALD have not been gathered and should be considered as NBS for X-ALD is expanded both in the U.S. and around the globe.

The psychosocial consequences of identification of females with X-ALD are understudied. In California, some mothers of these females reported distress about the lack of clinical follow-up, despite understanding that their daughters were not at-risk for pediatric symptoms [47]. For other conditions identified via NBS (cystic fibrosis and sickle cell hemoglobinopathy), parental perceptions of child vulnerability were significantly higher for patients identified as carriers than their age-matched peers [48]. Vulnerable child syndrome is a known complication of false-positive results of expanded NBS for biochemical genetic disorders [49,50].

In Japan, X-ALD diagnoses have long occurred through a streamlined diagnostic approach at a single referral center [51]. That system screened asymptomatic relatives of known X-ALD cases to facilitate presymptomatic diagnosis but ultimately relied on the identification of symptomatic individuals as a starting point [51,52]. In April 2021, a pilot study of NBS for X-ALD in Japan was initiated in which only results from babies with typical male-appearing external genitalia at birth were recorded [52]. Although rare, this presents an interesting challenge in cases of atypically appearing external genitalia or gonadal dysgenesis, both of which can obscure identification of babies with a single X chromosome. To avoid some of these issues while still permitting X-ALD NBS in male infants, the Netherlands enacted a screening protocol limited to males [53]. When a sample is identified that contains elevated C26:0-LPC, X chromosomes are “counted” using a commercially available PCR-based kit, thereby permitting exclusion of samples from female newborns from subsequent steps in the X-ALD screening process [53]. This approach has not been utilized in the U.S.

## 5. Conclusions

NBS for X-ALD promises early detection of at-risk patients and their family members. These results demonstrate that although newborn screening is a powerful tool, it is complicated by variant interpretation, lack of genotype/phenotype correlation, lack of biomarkers to predict disease course, and treatment that is more reactive than proactive. Our experience with the identification of a false-negative result also provides a cautionary lesson that no clinical test is perfect. As more babies across the U.S. are screened for X-ALD, additional work is needed to better characterize *ABCD1* variants, predict male patients who will demonstrate the cerebral form of the disease, and support an expanded population of heterozygous female patients.

## Figures and Tables

**Figure 1 IJNS-08-00024-f001:**
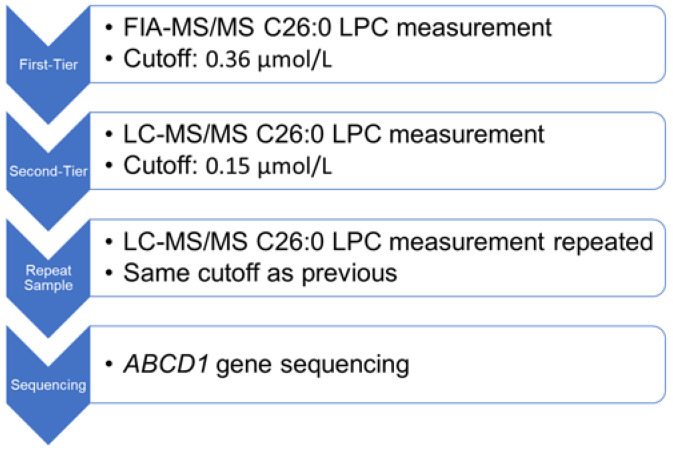
The Pennsylvania X-ALD newborn screening algorithm utilizes a tiered approach. C26:0-LPC is measured first with flow injection analysis tandem mass spectrometry (FIA-MS/MS), then with liquid chromatography tandem mass spectrometry (LC-MS/MS). Repeat dried blood spot samples were requested for newborns with concerning second-tier screens. Sequencing of the *ABCD1* gene was performed for repeat specimens with C26:0-LPC exceeding the cutoff value.

**Figure 2 IJNS-08-00024-f002:**
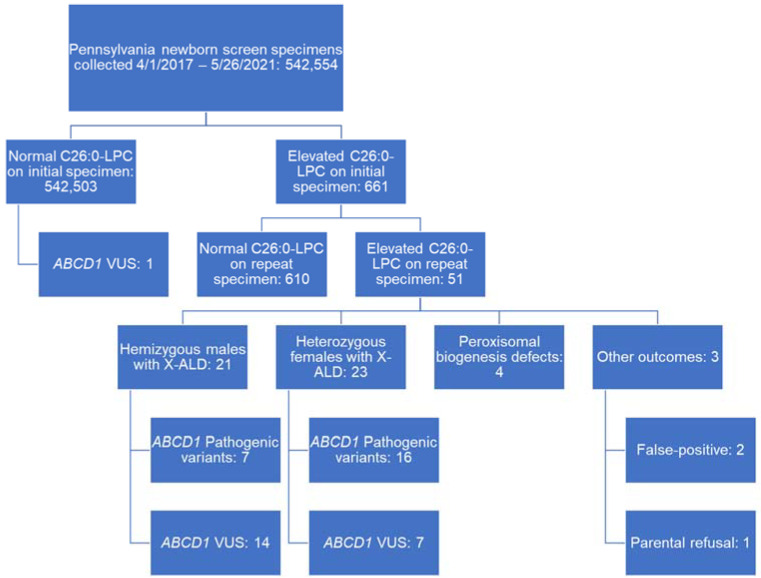
A visual summary of Pennsylvania X-ALD newborn screening results between 1 April 2017 and 26 May 2021. LPC = lysophosphatidylcholine; X-ALD = X-linked adrenoleukodystrophy; VUS = variant of uncertain significance.

**Table 1 IJNS-08-00024-t001:** Newborn screen, biochemical, and molecular features of 44 Pennsylvanian infants who screened positive for X-ALD between 1 April 2017 and 26 May 2021 and harbored *ABCD1* variants. Screening was considered positive for second-tier LC-MS/MS quantitation of C26:0-LPC > 0.15 µmol/L on initial and repeat dried blood spot specimens. First-tier results from FIA-MS/MS are not shown. Confirmatory biochemical testing with C26:0 quantitation was performed on a clinical basis (reference range: 0.17–0.73 µmol/L). “-” denotes that the clinical information was unavailable. All units are in µmol/L. *ABCD1* variants and their laboratory interpretation using ACMG criteria are listed. Variants identified in the ALD Variant Database are indicated as previously reported. Inheritance pattern is provided for families in whom parental testing was performed and available. Finally, known family history is denoted for patients in whom there were family members with confirmed, symptomatic X-ALD phenotype. “*” indicates individuals with family members identified via NBS who are known to carry the same variant but are asymptomatic. “^” indicates family individuals whose medical histories were suspicious for symptomatic individuals but for whom an X-ALD diagnosis had not been made (for example: an adult male uncle with progressive gait disturbance).

#	Sex	NBS C26:0-LPC Initial/Repeat		VLCFA C26:0	*ABCD1* Variant		Classification	Previously Reported?	Inheritance	Family History
1	Female	0.32	0.52	3.40	c.887A>C	p.Tyr296Ser	Likely pathogenic	Yes	Maternal	No
2	Female	0.38	0.3	2.12	c.1166G>A	p.Arg389His	Pathogenic	Yes	Paternal	No
3	Female	0.31	0.23	1.03	c.1415_1416delAG	p.Gln472Profs*84	Pathogenic	Yes	Maternal	Yes
4	Female	0.62	0.32	-	c.1447dupA	-	Pathogenic	No	-	-
5	Female	0.29	0.34	1.97	c.1516dupA	-	Pathogenic	No	De novo	No
6	Female	0.43	0.26	-	c.1628del	p.Pro543Argfs*15	Pathogenic	Yes	-	No
7	Female	0.31	0.28	2.34	c.1690delG	-	Pathogenic	No	De novo	No
8	Female	0.73	0.47	2.67	c.1978C>T	p.Arg660Trp	Pathogenic	Yes	Maternal	No ^
9	Female	0.54	0.4	1.84	c.2135G>A	p.Arg712His	Pathogenic	Yes	Paternal or De novo	No
10	Female	0.81	0.39	-	c.264C>A	p.Cys88*	Pathogenic	Yes	De novo	No
11	Female	0.58	0.37	0.93	c.346G>A	p.Gly116Arg	Pathogenic	Yes	Maternal	Yes
12	Female	0.39	0.35	2.52	c.390dupT	-	Pathogenic	No	Paternal or De novo	No
13	Female	0.51	0.29	2.50	c.521A>G	p.Tyr174Cys	Pathogenic	Yes	-	No
14	Female	0.4	0.29	2.22	c.838C>T	p.Arg280Cys	Pathogenic	Yes	-	No
15	Female	0.63	0.36	-	c.978G>A	p.Trp326*	Pathogenic	No	-	-
16	Female	0.48	0.37	3.48	Deletion of exons 3 and 4	-	Pathogenic	No	-	No
17	Female	0.49	0.26	2.99	c.1533C>G	p.Cys511Trp	VUS	Yes	Maternal	No ^
18	Female	0.55	0.22	1.39	c.262T>C	p.Cys88Arg	VUS	No	Maternal	No
19	Female	0.33	0.34	1.79	c.467G>A	p.Gly156Asp	VUS	No	Maternal	No
20	Female	0.28	0.28	1.46	c.700C>T	p.Arg234Cys	VUS	Yes	Maternal	No ^
21	Female	0.24	0.17	-	c.739G>A	p.Ala247Thr	VUS	Yes	Maternal	No *
22	Female	0.24	0.21	2.14	c.880G>A	p.Ala294Thr	VUS	Yes	Maternal	No
23	Female	0.23	0.19	0.76	c.970C>T	p.Arg324Cys	VUS	Yes	Maternal	No ^
24	Male	0.7	0.2	2.32	c.565C>T	p.Arg189Trp	Likely pathogenic	Yes	Maternal	Yes
25	Male	0.52	0.56	-	c.1390C>T	p.Arg464*	Pathogenic	Yes	Maternal	Yes
26	Male	0.84	0.9	-	c.1415_1416delAG	p.Gln472Profs*84	Pathogenic	Yes	Maternal	No
27	Male	1.08	0.52	2.90	c.1661G>A	p.Arg554His	Pathogenic	Yes	Maternal	Yes
28	Male	0.62	0.36	-	c.1772G>A	p.Arg591Gln	Pathogenic	Yes	Maternal	No *
29	Male	0.62	0.33	-	c.1772G>A	p.Arg591Gln	Pathogenic	Yes	Maternal	No *
30	Male	0.76	0.61	4.06	c.796G>A	p.Gly266Arg	Pathogenic	Yes	Maternal	No
31	Male	0.28	0.2	1.51	c.1184C>T	p.Ala395Val	VUS	No	Maternal	No *
32	Male	0.35	0.29	2.50	c.1253G>A	p.Arg418Gln	VUS	Yes	Maternal	No ^
33	Male	0.55	0.3	3.13	c.1448C>T	p.Thr483Met	VUS	No	Maternal	No ^
34	Male	0.64	0.75	-	c.1832A>G	p.Gln611Arg	VUS	Yes	-	-
35	Male	0.48	0.2	-	c.229_237delTGGCTCCTG	p.Trp77_Leu79del	VUS	Yes	-	-
36	Male	0.37	0.25	3.35	c.229_237delTGGCTCCTG	p.Trp77_Leu79del	VUS	Yes	Maternal	No ^
37	Male	0.3	0.27	2.60	c.452T>C	p.Ile151Thr	VUS	No	Maternal	Yes
38	Male	0.67	0.33	-	c.487C>T	p.Arg163Cys	VUS	No	-	-
39	Male	0.82	0.39	2.67	c.700C>T	p.Arg234Cys	VUS	Yes	Maternal	No *
40	Male	0.69	0.3	3.18	c.700C>T	p.Arg234Cys	VUS	Yes	Maternal	Yes
41	Male	0.29	0.27	-	c.739G>A	p.Ala247Thr	VUS	Yes	-	-
42	Male	0.86	0.97	3.05	c.824G>C	p.Arg275Pro	VUS	Yes	Maternal	No ^
43	Male	0.41	0.33	1.34	c.851C>T	p.Ser284Leu	VUS	No	Maternal	No
44	Male	0.31	0.27	-	c.851C>T	p.Ser284Leu	VUS	Yes	Maternal	No *

**Table 2 IJNS-08-00024-t002:** Newborn screen, biochemical, and molecular features of seven Pennsylvanian infants who screened positive for X-ALD between 1 April 2017 and 26 May 2021 and did not harbor *ABCD1* variants. Newborn screening was considered positive for second-tier LC-MS/MS quantitation of C26:0-LPC > 0.15 µmol/L on initial and repeat dried blood spot specimens. First-tier results from FIA-MS/MS are not shown. Confirmatory biochemical testing with C26:0 quantitation was performed on a clinical basis (reference range: 0.17–0.73 µmol/L). “-” denotes that the clinical information was unavailable. All units are in µmol/L.

#	Sex	NBS C26:0-LPC Initial/Repeat		VLCFA C26:0	*ABCD1* Variant	Screen Outcome
1	Female	0.64	0.4	2.57	none	Parent refusal
2	Female	0.36	0.29	0.76	none	False positive
3	Male	0.33	0.19	0.90	none	False positive
4	Female	0.69	0.48	2.47	none	Peroxisomal biogenesis defect
5	Male	1.32	1.36	8.75	none	Peroxisomal biogenesis defect
6	Male	1.19	1.82	21.48	none	Peroxisomal biogenesis defect
7	Unknown	2.18	2.22	-	none	Peroxisomal biogenesis defect

**Table 3 IJNS-08-00024-t003:** Comparison of newborn screening strategies for X-ALD across seven U.S. states. All states employed a two-tier strategy for identifying infants requiring biochemical genetics referrals. Sequencing of the *ABCD1* gene was performed either prior to the referral by the NBS laboratory as a “third-tier” screen or ordered clinically by the referral provider. FIA = flow injection analysis; LC = liquid chromatography; HPLC = high-performance liquid chromatography; MS/MS = tandem mass spectroscopy.

State	Tier 1: Method/Target/Cutoff		Tier 2: Method/Target/Cutoff		*ABCD1* Sequencing
California [43]	FIA-MS/MS	C26 ≥ 0.42 µmol/L	LC-MS/MS	C26 ≥ 0.22 µmol/L ^a^	Integrated with NBS
Georgia [42]	FIA-MS/MS	CLIR analysis of C20, C22, C24, and C26 LPC	LC-MS/MS	C26:0-LPC > 0.30 nmol/mL	Following referral
Illinois [44]	LC-MS/MS	Borderline: C26:0-LPC ≥ 0.18 µmol/L Positive: C26:0-LPC ≥ 0.28 µmol/L	LC-MS/MS	Borderline: C26:0-LPC ≥ 0.18 µmol/L Positive: C26:0-LPC ≥ 0.28 µmol/L	Following referral
New York [2]	MS/MS	C26:0-LPC	HPLC-MS/MS	C26:0-LPC	Integrated with NBS
North Carolina [41]	HPLC-MS/MS	C24:0-LPC ≥ 0.175 µmol/L and C26:0-LPC ≥ 0.08 µmol/L	Duplicate HPLC-MS/MS	Median C26:0-LPC ≥ 0.15 µmol/L OR Median C26:0-LPC 0.08–0.15 µmol/L and C24:0-LPC ≥ 0.175 µmol/L	Integrated with NBS
Minnesota [40]	LC-MS/MS	Borderline: C26:0-LPC ≥ 0.16 µmol/L Positive: C26:0-LPC ≥ 0.30 µmol/L	Repeat LC-MS/MS	C26:0-LPC ≥ 0.16 µmol/L	Following referral
Pennsylvania	FIA-MS/MS	C26:0-LPC > 0.36 µmol/L	LC-MS/MS	C26:0-LPC > 0.15 µmol/L	Integrated with NBS

^a^ California changed its second-tier cutoff value from ≥0.15 µmol/L to ≥0.22 µmol/L to improve test performance 22 months after NBS for X-ALD began [43].

**Table 4 IJNS-08-00024-t004:** Comparison of U.S. newborn screening outcomes for X-ALD across six publications plus the present study. Positive screens and how those were classified as various diagnoses or false-positive cases were defined by their respective authors.

State	Publication	Study Length	Total # Screened	Positive Screens	Male X-ALD	Female Heterozygote	Peroxisome Biogenesis Disorder	Other Genetic Syndrome
California	Matteson et al., 2021 [43]	4 years	1,854,631	355	95	110	23	12
Georgia	Hall et al., 2020 [42]	7 months	51,081	11	1	0	2	0
Illinois	Burton et al., 2022 [44]	1 year 11 months	276,000	34 ^a^	7	10 ^b^	3	0
New York	Moser et al., 2016 [2]	2 years 8 months	630,000	53	20 ^c^	22	- ^d^	- ^d^
North Carolina	Lee et al., 2020 [41]	6 months	52,301	12	3	3	1	1
Minnesota	Wiens et al., 2019 [40]	1 year	67,836	14	9	5	0	0
Pennsylvania	Present Study	4 years 2 months	542,554	51	21	23	4	0

^a^ Illinois also employs a system in which initial dried blood spot C26:0-LPC levels ≥0.28 µmol/L are considered positive and levels 0.18–0.28 µmol/L are considered borderline. Repeat dried blood spot specimens are requested for borderline cases and considered positive if C26:0-LPC levels are ≥0.28 µmol/L or borderline if C26:0-LPC levels are 0.18–0.28 µmol/L. Here, all positive screens are included, regardless of whether they were positive on the first or second dried blood spot specimen. ^b^ One female individual was found to be homozygous for her *ABCD1* variant due to isodisomy X and is included in this number. ^c^ One male individual was found to be heterozygous for his *ABCD1* variant due to 47,XXY and is included in this number. ^d^ Ten cases screened positive but did not harbor an identified *ABCD1* variant. It was included whether any of these cases represented alternative diagnoses.

## Data Availability

The data presented in this study are available on request from the corresponding author. The data are not publicly available due to privacy and ethics issues.

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
