# Peer review of "Newborn Screening for X-Linked Adrenoleukodystrophy: Review of Data and Outcomes in Pennsylvania"

_2409-515X, 2022, doi:10.3390/ijns8020024_

Round 1

Reviewer 1 Report

The manuscript by Dr. Priestley et al describes their experience with newborn screening for X-ALD in Pennsylvania. The authors report detailed information about the birth incidence, false positives, ABCD1 variants detected, C26:0 levels, and family cascade testing. They also compare their incidence data with that published so far from other states. The manuscript is well written with no obvious deficiencies. My comments are limited to minor typos.

Line 122: The adjective “denatured” is inappropriate for a lipid. Should be deleted.

Line 320: The comment about this report being the longest follow-up period to date, should be qualified to “longest reported” follow-up period to date, since NY has been screening longer than PA.

Line 345: typo “that” should be “than”.

Table 3: typo in North Carolina “C260-LPC” should be “C26:0-LPC”.

Author Response

Response to Reviewer 1 Comments

Point 1: Line 122: The adjective “denatured” is inappropriate for a lipid. Should be deleted.

Point 2: Line 320: The comment about this report being the longest follow-up period to date, should be qualified to “longest reported” follow-up period to date, since NY has been screening longer than PA.

Point 3: Line 345: typo “that” should be “than”.

Point 4: Table 3: typo in North Carolina “C260-LPC” should be “C26:0-LPC”.

Response to all points: We thank the reviewer for their attention to detail and have fixed these errors according to their suggestions.

Reviewer 2 Report

Thank you for this manuscript, it is thorough and the states comparison will be helpful.

Below is a list of suggestions for this paper's improvement.

  1. In first paragraph of the introduction the ALD Mutation Database is referenced, this has changed and is now called "The ALD Variant Database".
  2. In a couple of places in the introduction you make note that VLCFA are not elevated in females.  I believe that reference 17 states that diagnosis in females is improved when using the C26:0-LPC marker.  I suggest pointing this out.
  3. In second to last paragraph of intro, it says "Given the benefits of early disease identification on treatment options and disease outcomes in the adrenal insufficiency form... "  I think also true for treatment of CCALD form.
  4.  In results section, may be of interest for other programs to know the first tier screen positive rate in PA.
  5. For the single false negative, was this sample missed in first tier or both first and second tier? Is this what you discuss in lines 272-279?  This paragraph is not clear (lines 272-279).  In line 276, you mention our patient then in line 277 say "who was born outside of the study period.." not sure which brother is being referred to here.
  6. In results section, line 192 says twenty-two females, but in Figure 2 it says 23. 
  7. In section 3.2, four patients were reported with adrenal insufficiency and the associated cDNA variants were reported. For ease of use, can the case number be provided as well?
  8. Line 240 refers to the ALD Mutation Database, which has recently been named the ALD Variant Database.
  9. Table 4 provides publications, can this include the reference number?
  10. Also in Table 4, this may not be accurate, as  PBDs are inaccurate for at least one state.  Also, for false positives, the criteria for what's a false positive for each state is not explained. Are the criteria the same in each of the associated publications (are these apples to apples comparisons?) The FP rate for at least one state seems out of line with what has been reported at meetings.  Maybe you are inferring the the FP rates based on comparison of screen positives to male and female ALD cases?  Could it be that non-ALD cases are counted as FPs (such as Zellweger cases?).  I suggest removing the FP column unless you are certain these are correct (from my information, they are not).
  11. In Discussion Section as well as earlier in the manuscript a false negative is mentioned.  This was based on determination that a younger sib was picked up in screening but the older boy was missed.  Would be good to clearly point out which case is the younger sib case.  Did this child have a borderline result. How certain is the diagnosis of either of these siblings? Can more be said about their ABCD1 variant and the reported history of disease associated with this variant?
  12.  In Results section lines 342-350 talks about higher incidence in NBS and possibly attributing to this to mild variants and VOUS. If you look at marker concentrations in screening and diagnosis, do you see lower marker concentrations in these cohorts than those with Pathogenic variants and or family history?  Would be good to report if any correlation.

Author Response

Response to Reviewer 2 Comments

Point 1: In first paragraph of the introduction the ALD Mutation Database is referenced, this has changed and is now called "The ALD Variant Database".

Response 1: We thank the reviewer pointing this out. We have changed our wording to reflect this update.

Point 2: In a couple of places in the introduction you make note that VLCFA are not elevated in females. I believe that reference 17 states that diagnosis in females is improved when using the C26:0-LPC marker.  I suggest pointing this out.

Response 2: This is an important point. In lines 65-66 we wrote, “…females do not always demonstrate elevations in VLCFAs [17,18]…” and in lines 76-77 we write “…elevated plasma VLCFAs (which are not uniformly present in female heterozygotes [18])…” We have added a sentence clarifying the evidence to support these points.

Point 3: In second to last paragraph of intro, it says "Given the benefits of early disease identification on treatment options and disease outcomes in the adrenal insufficiency form... "  I think also true for treatment of CCALD form.

Response 3: Thank you for this suggestion. We have added the CCALD phentoype to this sentence.

Point 4: In results section, may be of interest for other programs to know the first tier screen positive rate in PA.

Response 4: This is an excellent suggestion and we have incorporated this number (661) in Figure 2 and the results section.  

Point 5: For the single false negative, was this sample missed in first tier or both first and second tier? Is this what you discuss in lines 272-279?  This paragraph is not clear (lines 272-279).  In line 276, you mention our patient then in line 277 say "who was born outside of the study period.." not sure which brother is being referred to here.

Response 5: We apologize for this lack of clarity. Because the sample was normal on first-tier testing, no second-tier testing was performed. The younger brother had a positive NBS and was born outside the study period. We have revised the paragraph to clarify and include these details. See below in Response to 11.

Point 6: In results section, line 192 says twenty-two females, but in Figure 2 it says 23.

Response 6: Thank you for this attention to detail. We have corrected this error such that both the text and the figure reflect the 23 females.

Point 7: In section 3.2, four patients were reported with adrenal insufficiency and the associated cDNA variants were reported. For ease of use, can the case number be provided as well?Response 7: Thank you for this excellent suggestion. We have included the case numbers in the text for both those patients and the other single patients discussed.

Point 8: Line 240 refers to the ALD Mutation Database, which has recently been named the ALD Variant Database.

Response 8: We have changed our wording to reflect this change. Thank you for bringing it to our attention.  

Point 9: Table 4 provides publications, can this include the reference number?

Response 9: This is also an excellent suggestion. The reference numbers are now included in the tables.

Point 10: Also in Table 4, this may not be accurate, as PBDs are inaccurate for at least one state.  Also, for false positives, the criteria for what's a false positive for each state is not explained. Are the criteria the same in each of the associated publications (are these apples to apples comparisons?) The FP rate for at least one state seems out of line with what has been reported at meetings.  Maybe you are inferring the the FP rates based on comparison of screen positives to male and female ALD cases?  Could it be that non-ALD cases are counted as FPs (such as Zellweger cases?).  I suggest removing the FP column unless you are certain these are correct (from my information, they are not).

Response 10: Thank you for this careful review of this table – these points are well-taken. We have reviewed the source data for the table, and included false positive cases as they were indicated or not by their study authors. This is now made explicit in the figure description. You are correct that New York simply listed 10 screen-positive cases as “no mutation” cases and we erroneously categorized these as false-positive cases. This has been amended and clarified as a footnote to the table. In doing so, we also clarified the screening results for Illinois (which reported both positive and borderline screens). We have also eliminated the false-positive column, as suggested.

Point 11: In Discussion Section as well as earlier in the manuscript a false negative is mentioned.  This was based on determination that a younger sib was picked up in screening but the older boy was missed.  Would be good to clearly point out which case is the younger sib case.  Did this child have a borderline result. How certain is the diagnosis of either of these siblings? Can more be said about their ABCD1 variant and the reported history of disease associated with this variant?

Response 11: Thank you for these suggestions. Based on this and Point 5, we have clarified our description of this case and provided additional details, as follows:

Additionally, one male child had a normal Pennsylvania NBS and was later found to carry an ABCD1 VUS. His initial NBS dried blood specimen had a C26:0-LPC level of 0.17 µM, well below the first-tier screening threshold of 0.36 µM. As a result, second-tier screening was not performed and a repeat dried blood spot specimen was not re-quested. He had a younger brother born three years later who had a Pennsylvania NBS concerning for elevated C26:0-LPC and was found to harbor an ABCD1 VUS (c.86C>A; p.Ala29Asp). This variant has not been reported in the ALD Variant Database [4]. Be-cause the younger brother was born after the conclusion of our study period, his posi-tive screen is not reflected as a case in Table 1. Cascade testing was performed on family members, including the older brother, who was found to harbor the same maternally inherited ABCD1 VUS. There was no family history of X-ALD diagnosis or suspicious neurologic phenotypes, although the maternal grandfather did have peripheral neu-ropathy believed to be a consequence of diabetes mellitus type II. The older brother was evaluated by a biochemical geneticist and VLCFAs were obtained. These showed an elevated C26:0 level of 1.34 µmol/L (reference range: 0.17-0.73 µmol/L). He is being followed by a multidisciplinary X-ALD team, including neurology and endocrinology. At 3 years of age, he is asymptomatic.

We note that Pennsylvania does have a formal borderline result screening category, but the older brother who screened negative was firmly below the screening cutoff and the younger brother who screened positive was firmly above the screening cutoff on both initial and repeat dried blood spot specimens (not shown because his positive screen occurred recently and was outside the timeframe for results review of this study thus not approved by our IRB).

Point 12: In Results section lines 342-350 talks about higher incidence in NBS and possibly attributing to this to mild variants and VOUS. If you look at marker concentrations in screening and diagnosis, do you see lower marker concentrations in these cohorts than those with Pathogenic variants and or family history?  Would be good to report if any correlation.

Response 12: Thank you for this suggestion and providing us the opportunity to clarify our language. We see lower marker concentrations in males with ABCD1 VUSs than males with likely pathogenic/pathogenic variants, but the trend is far from statistically significant. We have added this to our results and discussion sections. We have also clarified our discussion of the potential identification of milder variants.

Reviewer 3 Report

The authors describe the first four years’ experience of X-ALD newborn screening in Pennsylvania.  From 542k screened 51 screen positive babies were referred, these included 21 boys with x-ALD and 23 females.   They report an incidence of 1:13,000 among boys and 1:11,000 in girls.

They go on to compare this data and the approach to screening in six other states and taken together suggest an overall incidence of 1:11,000 for the condition.

The paper is interesting and very well written.  It needs only minor clarifications: 

The screening algorithm – Fig 1 on p3 includes ABCD1 gene sequencing and while gene sequencing may be initiated by the screening lab, it appears that the results of screening are not used to determine clinical referral which rests only upon the second C26-LPC result – see line 143 p4.   This is a little confusing and it may be wise to indicate the point of referral, related to C26.0 on Fig 1.

The narrative on p6 lines 222 – 225 is confusing: ‘There was no history of either X-ALD diagnosis or suspicious neurological phenotypes for 17/38 cases (45%).   There was no history of either X-ALD or suspicious neurological phenotype for an additional 6/38 cases (16%) despite other family members being identified as harbouring the ABCD1 variant.’   The distinction between the 17/38 cases and the additional 6/38 cases is unclear – I suspect that the wording is incorrect or needs modified to improve understanding.

With these revisions, I would recommend that the paper is published.

Author Response

Response to Reviewer 3 Comments

Point 1: The screening algorithm – Fig 1 on p3 includes ABCD1 gene sequencing and while gene sequencing may be initiated by the screening lab, it appears that the results of screening are not used to determine clinical referral which rests only upon the second C26-LPC result – see line 143 p4.   This is a little confusing and it may be wise to indicate the point of referral, related to C26.0 on Fig 1.

Response 1: This point is well-taken. We have clarified this point in the table legend and amended the table to indicate where ABCD1 sequencing occurs integrated with NBS and where it occurs following referral to a geneticist. Thank you for helping us make this more straight-forward to readers. 

Point 2: The narrative on p6 lines 222 – 225 is confusing: ‘There was no history of either X-ALD diagnosis or suspicious neurological phenotypes for 17/38 cases (45%).   There was no history of either X-ALD or suspicious neurological phenotype for an additional 6/38 cases (16%) despite other family members being identified as harbouring the ABCD1 variant.’   The distinction between the 17/38 cases and the additional 6/38 cases is unclear – I suspect that the wording is incorrect or needs modified to improve understanding.

Response 2: This point is also well-taken. We agree that this was not as clear as we would hope. While both groups lack family history of or concerning for X-ALD, the distinction is that the 6/38 group have asymptomatic family members with ABCD1 variants identified via NBS. We feel this is an interesting observation, and have clarified our varbiage accordingly. The section now reads:

“There was a known family history of X-ALD diagnosis for 7/38 cases (18%). Two of those variants were classified as VUS, one as likely pathogenic. Although there was no history of family members with formal X-ALD diagnoses, 8/38 cases (21%) had family histories concerning for neurological abnormalities that could be consistent with X-ALD phenotypes. Examples include adult female relatives with leg stiff-ness/weakness or unexplained incontinence, and adult male relatives with atypical multiple sclerosis, neuropathy, tremor, or gait abnormalities. In 6/38 cases (16%), there were family members with ABCD1 variants identified on NBS but no family history of X-ALD phenotypes, including in the screen-positive individuals. Two of those, male children from the same family, harbor pathogenic ABCD1 variants. Finally, there was no history of X-ALD diagnosis, positive X-ALD NBS, or suspicious neurological phe-notypes in the extended families for the remaining 17/38 cases (45%).”

We thank the reviewer for this opportunity to clarify our work.